# Enhancing air quality predictions in Chile: Integrating ARIMA and Artificial Neural Network models for Quintero and Coyhaique cities

Fidel Vallejo[1,2]*, Diana Yáñez[3], Patricia Viñán-Guerrero[4], Luis A. Díaz-Robles[2], Marcelo Oyaneder[2,5], Nicolás Reinoso[5], Luna Billartello[5], Andrea Espinoza-Pérez[6,7], Lorena Espinoza-Pérez[6,7], Ernesto Pino-Cortés[8]

1 Industrial Engineering, National University of Chimborazo, Riobamba, Ecuador, 2 Particulas Environmental Engineering and Management, Chile, 3 Agroindustrial Engineering, National University of Chimborazo, Riobamba, Ecuador, 4 Engineering Faculty, National University of Chimborazo, Riobamba, Ecuador, 5 Chemical Engineering Department, Faculty of Engineering, University of Santiago of Chile, Estación Central, Santiago, Chile, 6 Program for the Development of Sustainable Production Systems (PDSPS), Faculty of Engineering, University of Santiago of Chile, Estación Central, Santiago, Chile, 7 Industrial Engineering Department, Faculty of Engineering, University of Santiago of Chile, Estación Central, Santiago, Chile, 8 Escuela de Ingeniería Química, Pontificia Universidad Católica de Valparaíso, Valparaíso, Chile

* fidel.vallejo@unach.edu.ec

**Data Availability Statement:** that the raw data necessary to replicate the results of our study have been made available. Specifically, the data used to

## Abstract

In this comprehensive analysis of Chile's air quality dynamics spanning 2016 to 2021, the utilization of data from the National Air Quality Information System (SINCA) and its network of monitoring stations was undertaken. Quintero, Puchuncaví, and Coyhaique were the focal points of this study, with the primary objective being the construction of predictive models for sulfur dioxide ($SO_2$), fine particulate matter ($PM_{2.5}$), and coarse particulate matter ($PM_{10}$). A hybrid forecasting strategy was employed, integrating Autoregressive Integrated Moving Average (ARIMA) models with Artificial Neural Networks (ANN), incorporating external covariates such as wind speed and direction to enhance prediction accuracy. Vital monitoring stations, including Quintero, Ventanas, Coyhaique I, and Coyhaique II, played a pivotal role in data collection and model development. Emphasis on industrial and residential zones highlighted the significance of discerning pollutant origins and the influence of wind direction on concentration measurements. Geographical and climatic factors, notably in Coyhaique, revealed a seasonal stagnation effect due to topography and low winter temperatures, contributing to heightened pollution levels. Model performance underwent meticulous evaluation, utilizing metrics such as the Akaike Information Criterion (AIC), Ljung-Box statistical tests, and diverse statistical indicators. The hybrid ARIMA-ANN models demonstrated strong predictive capabilities, boasting an $R^2$ exceeding 0.90. The outcomes underscored the imperative for tailored strategies in air quality management, recognizing the intricate interplay of environmental factors. Additionally, the adaptability and precision of neural network models were highlighted, showcasing the potential of advanced technologies in refining air quality forecasts. The findings reveal that geographical and climatic

generate the graphs in our manuscript have been uploaded to a publicly accessible repository on Kaggle (10.34740/kaggle/ds/4949501). Additionally, the data from the monitoring stations utilized in our study are publicly available on the website https://sinca.mma.gob.cl/.

**Funding:** FV received research funding as Director of the project Industrial Optimization by Mathematical Methods (OPRIM) at the National University of Chimborazo (UNACH). The funds were allocated by the Vice Presidency of Research for data analysis, manuscript writing, and covering the Article Processing Charges (APC). More information is available at https://www.unach.edu.ec/vicerrectorado-de-investigacionvinculacion-y-posgrado-ele/. Additionally, AE-P received research support from the Dirección de Investigación Científica y Tecnológica (DICYT) under grant 062317EP_Ayudante to facilitate data collection and experimental work. The funders had no role in study design, data collection and analysis, decision to publish, or preparation of the manuscript.

**Competing interests:** The authors have declared that no competing interests exist.

factors, especially in Coyhaique, contribute to elevated pollution levels due to seasonal stagnation and low winter temperatures. These results underscore the need for tailored air quality management strategies and highlight the potential of advanced modeling techniques to improve future air quality forecasts and deepen the understanding of environmental challenges in Chile.

## Introduction

The rapid surge in global population growth and the swift pace of industrialization on a global scale have ushered in a myriad of environmental challenges, with the deterioration of air quality standing out prominently [1]. It is especially acute in developing nations, where the burgeoning industrial activities often outpace environmental regulations and infrastructure [2]. Notably, countries such as India and China have garnered notoriety for their alarmingly high levels of air pollution, exemplified by annual fine particulate matter ($PM_{2.5}$) concentrations reaching 85 μg/m$^3$ in Delhi and 34.4 μg/m$^3$ in Beijing [3, 4]. It underscores the severe impact of industrialization and urbanization on air quality, posing substantial risks to public health [5]. Moreover, the global concern extends to low- and middle-income countries, where air quality in less than 1% of cities meets the air quality thresholds recommended by the World Health Organization [6]. It highlights a pervasive and critical issue that demands urgent attention and robust interventions to safeguard the health and well-being of populations. Shifting our focus to Latin America and the Caribbean, Chile is a notable case, standing out as a significant contributor to the region's air pollution landscape. In both 2021 and 2022, Chile accounted for a substantial proportion of 60 and 66.7% of the 15 most polluted cities in the region, surpassing WHO limits for $PM_{2.5}$ by at least sevenfold [7]. It emphasizes the severity of air quality challenges in the country, necessitating comprehensive strategies for mitigation and control.

Furthermore, the significance of sulfur dioxide ($SO_2$) emissions comes to the forefront, mainly attributable to Codelco, Chile's state-owned enterprise and the world's largest copper producer [8]. The production process, particularly in copper smelters, contributes significantly to $SO_2$ emissions, a noxious gas with detrimental health effects [9]. Understanding and addressing this aspect is crucial for comprehensive air quality management. Chile's geographical considerations add a layer of complexity and play a pivotal role in exacerbating air quality challenges [10]. Most Chilean cities are nestled in valleys due to the Andes Mountain Range and the Coastal Mountain Range that span the country. This unique topography restricts the dispersion of pollutants, and the situation is further compounded by frequent thermal inversion events during the winter months [11]. The intricate interplay of geographic factors and industrial activities underscores the need for tailored and effective air quality management strategies in Chile, considering both industrial processes and the distinct environmental features of the region.

Significant examples of Chilean cities facing these air quality challenges are Coyhaique, Quintero, and Puchuncaví. Annual mean $PM_{2.5}$ levels in southern Chilean cities have been significantly higher than WHO guidelines since 2017 [7]. Among these, Coyhaique is the most polluted city in the whole continent, with an average annual concentration of $PM_{2.5}$ corresponding to 37 μg/m$^3$ between 2017 and 2022. According to WHO, this value should not exceed five μg/m$^3$. It was declared a polluted zone for $PM_{10}$ in 2012 and $PM_{2.5}$ in 2016 [12, 13]. The main source of $PM_{10}$ and $PM_{2.5}$ emissions comes from the residential combustion of

wood; 96% of housing use firewood for heating and cooking food [14], contributing 99.9% of total $PM_{10}$ emissions and 99.67% of total $PM_{2.5}$ emissions [15, 16]. Critical pollution episodes by particulate matter are characterized by the limited dispersion capacity of pollutants in the basin during autumn and winter when winds average 2 m/s, temperatures range between -10˚C and 5˚C, and a mean relative humidity of 66.2% [17–19]. Coyhaique is situated in a valley in the Aysén Region of southern Chile, surrounded by the Andes Mountains. It has a cold oceanic climate characterized by cold winters, frequent snowfall, and mild summers. The region experiences significant temperature variations, with winter temperatures often dropping below freezing. The topography and climatic conditions contribute to frequent thermal inversions during autumn and winter, trapping pollutants close to the ground and leading to high concentrations of particulate matter [14].

Located in the northern coastal sector of the Valparaíso Region, Quintero and Puchuncaví host the "Ventanas Industrial Complex," which includes over 17 industrial facilities such as thermoelectric plants, petrochemical refineries, and a copper smelter. The climate is Mediterranean, with dry summers and mild, wet winters. The coastal location results in moderate temperatures and high humidity with prevalent sea breezes that can influence the dispersion of pollutants. However, atmospheric conditions sometimes lead to pollutant accumulation, affecting air quality in the region [20]. 2018 marked a dire incident in this region when approximately 1,415 individuals experienced poisoning from various gases, including methyl chloroform, nitrobenzene, toluene, sulfur dioxide, and particulate matter [9]. Particularly alarming was the disproportionate impact on school-going children from institutions such as "La Greda," "Alonso Quintero," and "Francia," prompting the National Disaster Prevention and Response Service (ONEMI) to declare a yellow alert for one week in the communes of Quintero and Puchuncaví. Similar incidents had been documented in 2011, indicating a recurring and persistent challenge in this industrialized zone [20]. The Ventanas Industrial Complex assumes a substantial role in Chile's emission profile, contributing to 22% of the nation's total emissions of carbon dioxide, particulate matter, sulfur dioxide, and nitrogen oxides. It makes it the second-largest sacrifice zone in terms of its impact on air pollution in the country, with Mejillones municipality claiming the first position at 32% [21]. The severity of the situation underscores the urgent need for stringent environmental regulations, effective monitoring, and proactive measures to mitigate the adverse effects of industrial activities on the air quality and public health in this critical zone.

Both studied zones grapple with the intricate interplay of natural features, with Coyhaique's mountainous surroundings influencing pollutant dispersion [19] and Quintero's and Puchuncavi's coastal proximity [22, 23]. This complexity underscores the urgency for coordinated efforts by the government and private sector, involving stringent regulations, effective enforcement, promotion of clean energy, and comprehensive public awareness campaigns.

In this context, providing alert systems in the short term becomes crucial to anticipate alert, pre-emergency, and environmental emergency levels with the final aim of mitigating the health impacts on the population [24]. Proactive measures can be enacted, such as halting industrial operations and reducing vehicular traffic in urban centers for specified durations [25]. Consequently, forecasting the concentration values of particulate or gaseous pollutants is indispensable for timely risk prediction.

Previous studies on air quality forecasting in Chile have provided valuable insights into the challenges and opportunities associated with predicting pollutant levels. Notably, Díaz-Robles et al. (2008) employed a hybrid model combining AutoRegressive Integrated Moving Average (ARIMA) and Artificial Neural Networks (ANN) for $PM_{10}$ forecasting in Temuco [11], showcasing improved accuracy over individual models. It was demonstrated that incorporating variables such as wind speed, precipitation, relative humidity, solar radiation, and atmospheric

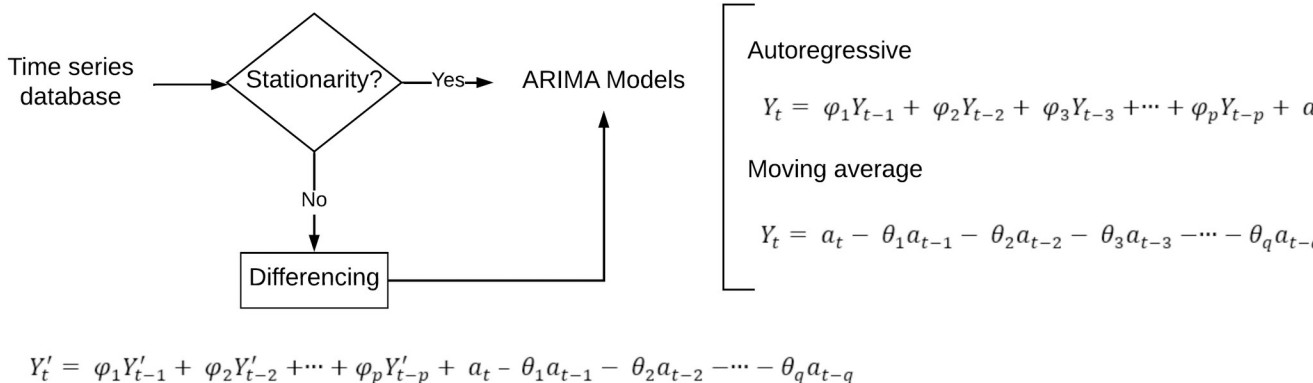

**Fig 1. ARIMA model setup, φ, θ are parameters of the prediction model for $y_t$.**

pressure enhanced the model's predictive capabilities. The study effectively captured complex patterns, achieving 100% accuracy in alert episodes and 80% in pre-emergency episodes. Additionally, a dedicated effort focused on Coyhaique by [17] developed an ANN model and a linear model for $PM_{2.5}$ forecasting, demonstrating the significance of accurate predictions in a region heavily impacted by wood stove emissions during fall-winter seasons. In this case, including meteorological variables like average temperature, wind speed, thermal amplitude, wind direction, and accumulated precipitation further contributed to precise $PM_{2.5}$ predictions. The results highlighted the neural network model's ability to achieve a Pearson correlation ($R^2$) of about 0.95, a normalized mean error of 18%, and an 84% prediction accuracy for critical air quality days in Coyhaique. In the context of pollution and socioeconomic variables, using multiple linear regression with ordinary least squares (OLS) presented insights into the relationship between air pollution and factors such as income poverty, multidimensional poverty, and energy poverty [26]. The study found significant positive correlations between air pollutants ($PM_{10}$, $PM_{2.5}$, and $SO_2$) and variables reflecting poverty levels by employing fixed effects for years and months.

Considering the limited utilization of statistical models for air quality assessment in Chile, the proposed study aims to build upon these foundations. The selection of ARIMA and ANN methodologies is justified by their proven effectiveness in capturing linear and nonlinear patterns in data from air quality. By building upon the strengths demonstrated in the cited works, the hybrid ARIMA-ANN approach is anticipated to enhance forecast accuracy, especially for critical pollutants such as $SO_2$, $PM_{2.5}$, and $PM_{10}$ in Chilean cities Quintero and Coyhaique.

Regarding predictive models, both ARIMA and ANN models serve as prominent methodologies for air quality forecasting, each with distinct advantages and limitations [27, 28]. ARIMA models offer simplicity in implementation and interpretation due to their linear nature. They effectively capture recurring patterns within time series data, a valuable trait for air quality forecasts. Additionally, these models demonstrate robustness in handling missing data and outliers [29]. However, ARIMA models have limitations. Before delving into the process depicted in Fig 1, ensuring the time series data's stationary is imperative, which is essential for ARIMA's functionality.

Furthermore, they might struggle when dealing with highly nonlinear or complex relationships in specific datasets. The application process involves several steps. First, the time series must be rendered stationary, involving the removal of trends and seasonality. Then, determining the order of each part of the ARIMA model is crucial for forecasting pollutant concentrations [30].

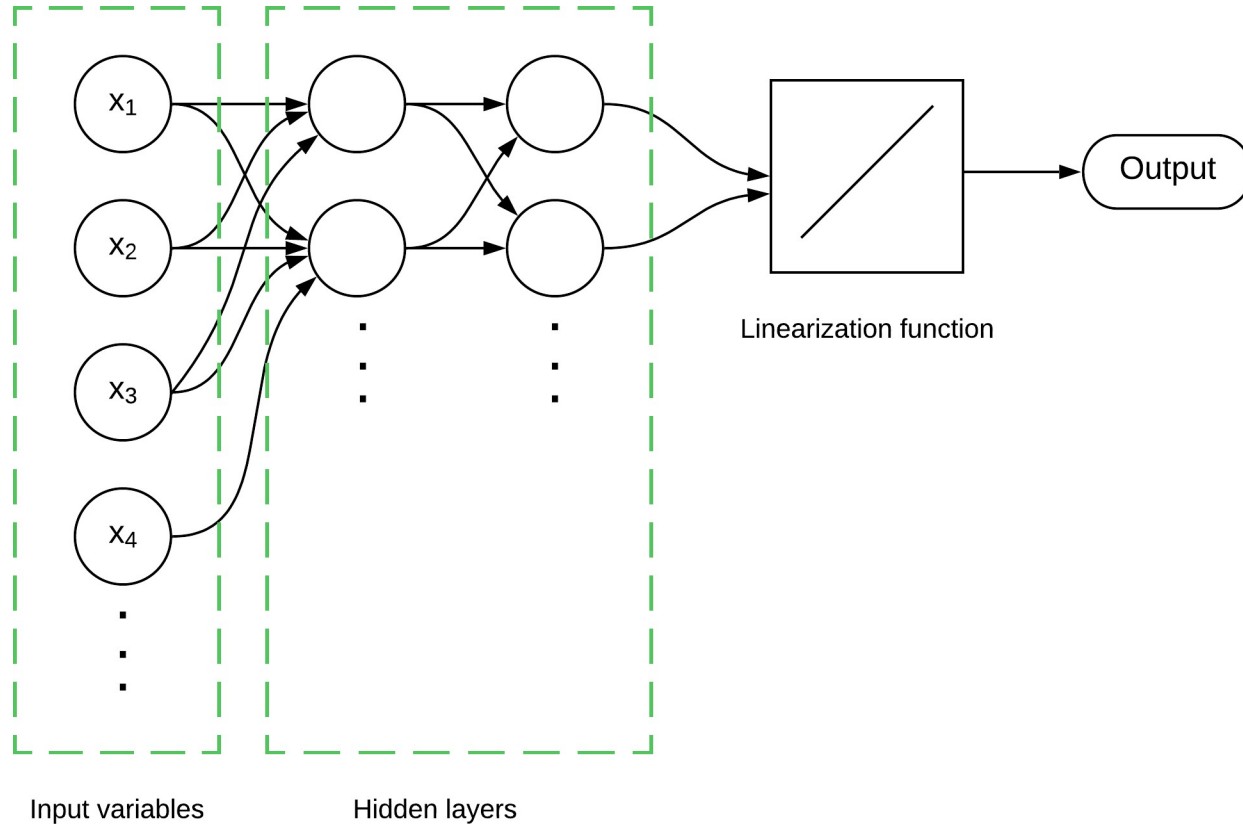

**Fig 2. Schematic representation of an ANN model.**

On the contrary, ANN models represent a category of nonlinear models deeply rooted in machine learning principles, mirroring the learning processes of real neurons [31]. The schematic illustration presented in Fig 2 provides a comprehensive overview of the intricate architecture inherent in an ANN model. This visual representation effectively captures the model's complexity by delineating the inputs, hidden layers, and outputs. Each input node corresponds to the initial parameters and variables input into the network, while the hidden layers epitomize sophisticated internal processing mechanisms. The output node serves as the endpoint, showcasing the final result generated by the model, revealing the intricate interconnections and weight distributions.

Significantly, the ANN algorithm autonomously adjusts these parameters to optimize error minimization, as [32] highlighted. An exemplary application of this model type is found in recurrent neural networks, which are particularly well-suited for time-series data analysis, a critical aspect in air quality forecasting. Leveraging past data, these models can make precise predictions about future air quality conditions.

Moreover, incorporating hyperparameter optimization techniques, such as the Keras tuner framework, is crucial in refining these models. This process involves fine-tuning various parameters to minimize the mean absolute percentage error (MAPE) metric, thereby further enhancing the accuracy of air quality forecasting models [27]. The amalgamation of cutting-edge technology and sophisticated machine learning models signifies a notable advancement and holds substantial promise in significantly elevating the accuracy of air quality forecasting. This technological leap is paramount for effective environmental monitoring and the success of public health initiatives, as underscored by contemporary research findings [33]. Despite

the strengths demonstrated by the ARIMA and ANN models in air quality forecasting, several limitations must be acknowledged. Data availability and quality pose significant challenges; gaps or inconsistencies in the monitoring data may occur due to equipment malfunctions or maintenance periods, potentially affecting model accuracy.

Additionally, air quality is influenced by many factors, including meteorological conditions, geographical features, and human activities [34]. While the models can incorporate key variables such as wind speed and direction, other influential factors like solar radiation, atmospheric pressure, and human-induced emissions could not be included due to data limitations. This exclusion may limit the models' ability to capture the full complexity of pollutant behavior.

Furthermore, the inherent limitations of the ARIMA and ANN models should be considered. Although effective in capturing linear patterns, ARIMA models may struggle with highly nonlinear or complex relationships in the data. While adept at modeling nonlinear patterns, ANN models require large datasets for training and can be prone to overfitting. The hybrid ARIMA-ANN approach aims to mitigate these individual limitations, but further refinement is possible. Additionally, focusing on specific regions like Quintero and Coyhaique with unique geographical and climatic conditions may limit the generalizability of the results to other areas. Extending the study to include more diverse locations and incorporating additional variables could enhance the robustness and applicability of the findings.

## Materials and methods

### Data collection

Chile maintains a network of 219 monitoring stations for meteorological variables and air quality, integrated into the "National Air Quality Information System" (SINCA) [35]. Therefore, the study's main goal was to propose models for each city capable of describing the main meteorological and air quality variables for an adequate forecast of $SO_2$, $PM_{2.5}$, and $PM_{10}$ levels. The Quintero station was chosen due to its proximity to populated residential areas, making it representative of human exposure to air pollutants resulting from both industrial emissions and urban activities. The Ventanas station was selected because it is located closest to the primary industrial emission sources in the area, such as copper smelters and coal-fired power plants. This station captures the direct impact of industrial activities on pollutant concentrations, providing valuable data on emissions from the industrial zone. In the case of Coyhaique, the monitoring stations selected are Coyhaique I and Coyhaique II. These are the only official monitoring stations in the Coyhaique region that provide continuous and validated air quality data. Their selection was essential to capture the air quality influenced by residential heating, especially during colder months when wood burning is prevalent. These stations offer comprehensive coverage of the area's air quality conditions, allowing for accurate modeling of pollutants like $PM_{2.5}$ and $PM_{10}$, which are significantly affected by local heating practices. As shown in Fig 3, the red dots represent the locations of these monitoring stations, illustrating their strategic placement concerning emission sources and populated areas.

The data extracted from SINCA corresponds to hourly data validated by the Ministry of the Environment for $SO_2$, $PM_{10}$, and $PM_{2.5}$ concentrations. The period analyzed corresponds to the years between 2016 and 2021—the monitoring stations operated by SINCA use internationally recognized methods and calibrated instruments for measuring air pollutants. Regular maintenance and calibration of equipment are performed to comply with quality assurance protocols established by the Chilean Ministry of the Environment. The data undergo rigorous validation processes, including checks for completeness, consistency, and plausibility, ensuring that only high-quality data are used for analysis [35].

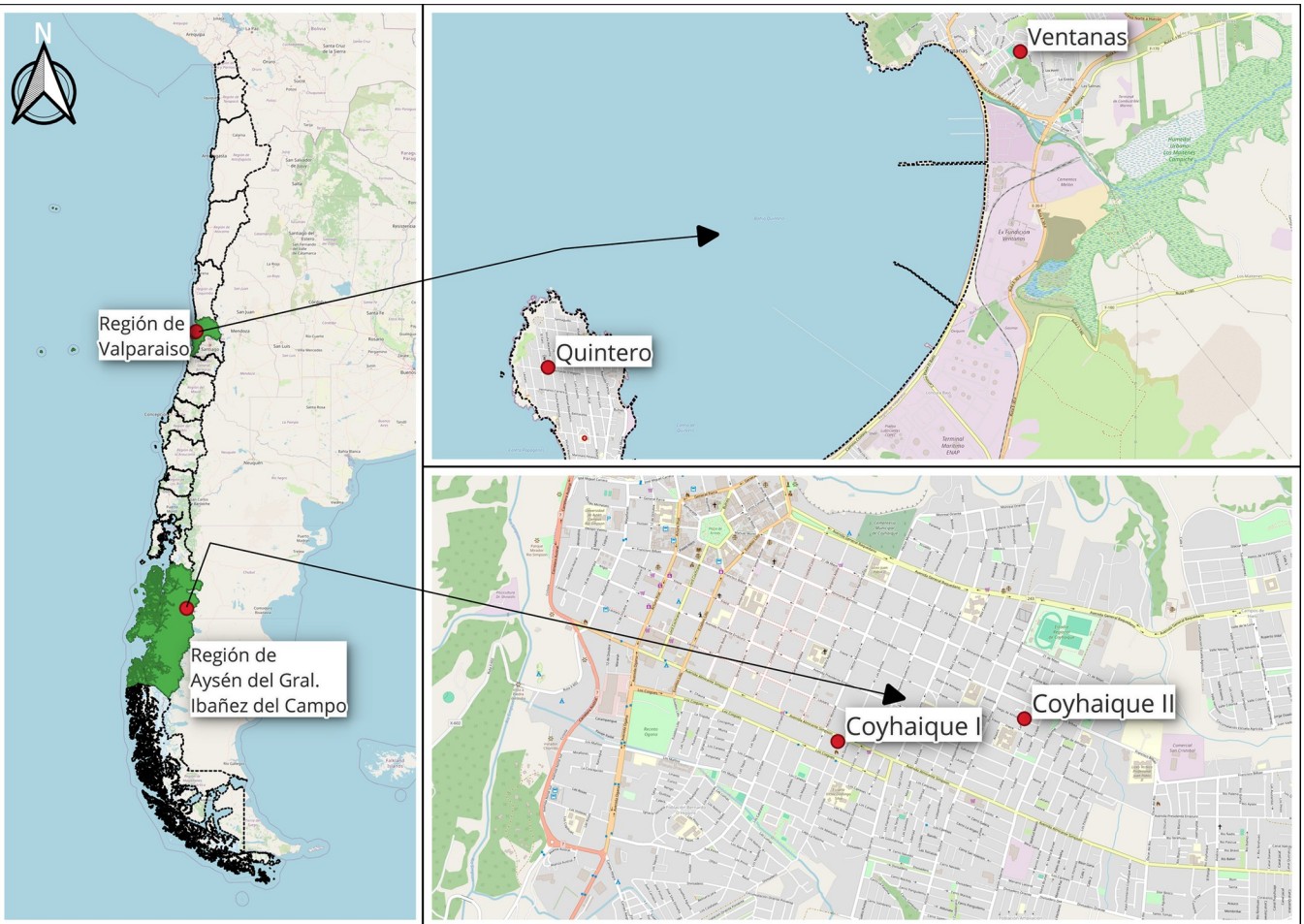

**Fig 3. Location map of Quintero, Puchuncaví, and Coyhaique monitoring stations.**

## Software and validation

The forecast was made using a hybrid ARIMA-ANN approach. The forecast [36] and caret packages [37, 38] of the R software were used for the generation, implementation, and validation. On the other hand, the analysis of seasonal cycles and identification of patterns was also carried out with OpenAir [39].

Both models were generated for the representative months of winter (June) and summer (December), that is, eight models. The first step was the determination of temporal and statistical patterns in the data, which were divided into the training and validation sets of the models (80–20%).

The generation of the ARIMA model and the parameter estimation followed the methodology presented in [11], using the *auto.arima()* function [36], which determines the order of the autoregressive (p), differencing (d), and moving average (q) parameters. The series was plotted before inputting the data into the function, and any anomalies were identified. Also, the data were processed to stabilize the variance with a logarithmic transformation. The procedure stopped when the first insignificant result was obtained. The parameter 'd' was chosen based on successive unit root tests of KPSS [40]. The data were tested for a unit root; if the test result was significant, the differenced data were tested for a unit root, and so on. Parameters 'p' and

'q' were selected based on minimizing the Akaike Information Criterion (AIC). This function performed the fewest possible differentiations for prediction purposes, as choosing parameter 'd' based on AIC minimization could lead to over-differentiation, affecting the forecast and widening prediction intervals.

On the other hand, the neural network model selected was the Neural Network Auto-Regressive (NNAR) algorithm, using the *nnetar()* function from the *forecast* package [36], which combines multi-layer neural network models with an autoregressive linear model. for better processing of information and working with complex dynamic systems. These models have only one hidden layer [36, 41], where the order 'k' denotes the number of neurons present. An order 'p' within the model indicates the autoregression order. An NNAR (p,0) model is similar to an ARIMA (p,0) model but without parameter limitations that ensure stationarity. For this reason, the same p values provided by the ARIMA model have been used. For the parameter *k*, the default value, which is half the number of input nodes (including external regressors, if present) plus 1, was evaluated [36, 42]. The gridSearch method [40] was also employed, creating a list of values and evaluating the model for each combination. The optimal model was selected based on statistical deviations. A total of 20 repetitions were conducted, corresponding to the number of networks fitted with different random starting weights. These networks were then averaged to generate the final forecasts.

For the validation, the Akaike Information Criterion (AIC) was used, which indicates the difference between the complexity of the model and its goodness of fit, measuring how explanatory and predictive a model is. When comparing this indicator between different models, it should be as small as possible since it would be closer to the complexity of the data, thus predicting values that are similar to those observed in reality [43].

Moreover, a comprehensive analysis of the residual values becomes imperative post-model development, as this subset might encompass crucial values necessitating model modifications to avert potential anomalies. The Ljung-Box statistical test emerges as a pivotal tool in this evaluation process, subjecting the residuals to three fundamental criteria that define a model's efficacy [44, 45]. Firstly, the test scrutinizes whether the time series of residuals aligns with the characteristics of white noise, denoting an absence of discernible trends or patterns. Secondly, it verifies the insignificance of residual lags, ensuring that past residuals do not exert undue influence on the current study period. Thirdly, the test assesses whether the distribution of residual values conforms to a normal distribution, contributing to the robustness of the model evaluation process [44].

The Ljung-Box test hinges on two statistical parameters, denoted as p and Q. The former (p) is associated with a null hypothesis, presupposing that the residuals exhibit characteristics of white noise. In contrast, the alternative hypothesis posits the absence of such characteristics. Conversely, the Q statistic is juxtaposed against a Chi-square distribution and must fall below the critical value in this distribution for the null hypothesis to hold. Furthermore, a p-value exceeding 0.05 is pivotal for affirming the fulfillment of the stipulated conditions. In comparing prediction models, the preferable choice is the one manifesting a lower value in the Q test statistic while concurrently satisfying the stipulated p criteria.

Several strategies were employed to prevent overfitting in our models. For the ANN models, regularization techniques such as L2 regularization and dropout layers were used to constrain model complexity. Early stopping was also implemented, monitoring the validation loss to halt training when necessary. Cross-validation techniques, specifically k-fold cross-validation, were employed to assess the model's performance on unseen data, ensuring its generalizability. The models were evaluated using performance metrics, including correlation coefficient ($R^2$), Root Mean Square Error (RMSE), and Mean Absolute Percentage Error (MAPE) on both training

and validation datasets. Residual analysis was conducted to verify the randomness of residuals, ensuring that the models were not overfitted. The p and Q statistics were pivotal in selecting the ARIMA models and developing the ANN models. Subsequently, the statistical values assumed a crucial role in evaluating the forecast performance of the hybrid ARIMA-ANN models, such as correlation coefficient ($R^2$), RMSE, and MAPE. We ensured a robust and accurate model evaluation process by integrating these performance metrics into the overfitting prevention strategies [30, 46].

# Results

## Statistical analysis of meteorological data

Over the period from 2016 to 2021, the data available for $PM_{2.5}$ and $PM_{10}$ at the Coyhaique I monitoring station stood at 98.2% and 97.9%, respectively, while for Coyhaique II, these figures were 96.5% for $PM_{2.5}$ and 96.0% for $PM_{10}$. On the other hand, Quintero stations showed between 96.2% and 98.8% of data availability. In Ventanas, the $PM_{2.5}$ ranged between 15 and 20 μg/m$^3$. The $SO_2$ had average peaks in winter above 18 μg/m$^3$ and minimums of 12 μg/m$^3$ in summer. PM2.5 had similar values at Quintero Station, but the $SO_2$ reached 70 μg/m$^3$ peaks, mainly in June.

The peak concentrations of $PM_{2.5}$ in Coyhaique I and Coyhaique II were observed during May, June, and July by examining the seasonal patterns. Notably, May registered the highest levels with 204.0 and 192.0 μg/m$^3$ for Coyhaique I and Coyhaique II, respectively. June and July followed suit, with concentrations ranging from 117.0 to 129.0 μg/m$^3$. Shifting the focus to $PM_{10}$ concentrations, a similar trend emerged, with the highest levels observed during May, June, and July. May recorded the highest concentrations for both monitoring stations, ranging from 157.0 to 222.0 μg/m$^3$. Interestingly, the summer months of December, January, and February displayed lower pollutant levels. December recorded the lowest concentrations, with 6.53 for $PM_{2.5}$ and 10.6 μg/m$^3$ for $PM_{10}$. January and February continued this trend, with concentrations ranging from 5.82 to 11.2 μg/m$^3$.

Geographical and climatic factors influence the challenging air quality conditions in Coyhaique. The basin's topography, combined with the seasonal prevalence of low winds during fall and winter, creates a stagnation effect that impedes the effective dispersion of pollutants [47]. The geographical context, characterized by surrounding hills and valleys, exacerbates this situation, accumulating pollutants in the air.

The harsh winter temperatures, often falling within the range of -10 to 5°C, further compound the problem. The necessity for heating during this period contributes significantly to the heightened levels of particulate matter. The increased use of combustion-based heating sources, such as wood-burning stoves, releases substantial amounts of pollutants into the atmosphere.

Several studies conducted in similar regions with complex topography and climatic conditions echo the challenges faced by Coyhaique. These studies highlight the intricate interplay between meteorological factors and air quality. The findings underscore the need for region-specific strategies to mitigate air pollution, acknowledging the unique environmental dynamics contributing to pollutant accumulation [48].

Comprehensive air quality management plans must consider regulatory measures, community engagement, and awareness to address these challenges. Implementing alternative heating technologies, promoting energy-efficient practices, and fostering a community-wide commitment to reducing emissions are essential components of a holistic approach [22, 49].

## Results of ARIMA models

The different prediction models generated are shown in Table 1, with their respective order, AIC and Ljung–Box test results for each pollutant, monitoring station, and study period.

**Model analysis for Quintero.** As mentioned above, the monitoring stations are located near the industrial zone, which is the primary source of air pollution; therefore, it was essential to consider where the toxic gases measured by the monitoring stations originate to determine the maximum concentrations of pollutants. It was directly affected by the direction of the wind, as the masses of toxic gases moved from the emission source to the monitoring station.

According to Table 1, the AIC decreased when another variable was added to the model generation, and many external factors changed the pollutant concentration measurement. It generated a more realistic picture of what is happening in the atmosphere, thus achieving better models and subsequent predictions. Considering the AIC values, the ARIMA models containing the average wind speed and the wind direction of interest as the external variables resulted in the lowest values.

On the other hand, the ARIMA above models meet the Ljung–Box test conditions, with p below 0.05. The following discussion considered the ARIMA models with those conditions. The order of the models depends on the study period at the Quintero Industrial Zone.

Considering $SO_2$ at Ventanas station for the December 2019 period, this model was affected only by the autoregression part of the ARIMA model. In other words, observations from previous periods of the same variable were highly influential in predicting pollutant concentrations. In this case, the order of the autoregression was 5, which indicates that the variable to be predicted was affected by observations of the same variable from five periods before the one under study; concerning the second-order shown, (d equal to 0) indicates that the time series under study did not need to be differentiated since the series was already stationary. As it was a model with a relatively high autoregressive order, its errors increase since the prediction generated depends on periods far from the one under study, thus increasing the difference between the observed and the prediction, causing the trend to be lost and forecasts outside the observed range.

By analyzing the same pollutant for the same period but for the Quintero station, the results are impacted by the autoregression sections and the moving average of the ARIMA model, the latter being the one that predominates in the predictions generated. In this case, the forecast is influenced by observing the same variable before the one under study (p = 1), which is also evident during the winter at the same station and for both $PM_{2.5}$ and $SO_2$ at the Ventanas station.

On the other hand, the ARIMA results with a value of q = 2 imply that the model is affected by the residuals of the observations of the same variable from two periods before the one under study. It was manifested for $SO_2$ for both periods at the Quintero station and $PM_{2.5}$ but for the December period at the Ventanas station.

It is also observed that we have a "d" equal to 0, which indicates that the time series does not need differentiation to be stationary. $SO_2$ concentrations were expressed for both stations in the summertime (December) and $PM_{2.5}$ at the Ventanas station in June.

ARIMA models with configurations (0,1,2), (1,1,1), (1,1,2), (1,0,2), and (1,0,3) have low order. A good prediction and low errors are expected, mainly because the forecast depends on periods that are not thus far from the current one. In contrast, model (5,0,0) is considered to have a relatively high autoregressive order, generating an increase in its errors since the prediction generated depends on periods far from the one under study, thus increasing the difference between the observed and the prediction, causing the trend to be lost and forecasts to be outside the observed range.

**Table 1. Statistical results for ARIMA models.**

| Period | Monitoring station | Pollutant | ARIMA (p,d,q) | Covariable | AIC | Test Ljung-Box | |
|---|---|---|---|---|---|---|---|
| | | | | | | Q | p |
| June-December | Coyhaique I | $PM_{2.5}$ | 3,1,1 | - | 2427.2 | 4.20 | 0.65 |
| | | | 2,1,1 | RH | 2217.0 | 4.56 | 0.33 |
| | | | 5,1,0 | $PM_{10}$ | 1748.0 | 58.00 | <0.01 |
| | | | 5,1,0 | RH, $PM_{10}$ | 1462.8 | 39.06 | <0.01 |
| | | | 5,1,0 | ws | 2327.5 | 37.88 | <0.01 |
| | | $PM_{10}$ | 5,1,0 | - | 64.7 | 34.24 | <0.01 |
| | | | 5,1,0 | RH | 85.9 | 29.79 | <0.01 |
| | | | 3,0,0 | $PM_{2.5}$ | -673.0 | 164.3 | <0.01 |
| | | | 1,0,0 | RH, $PM_{2.5}$ | -582.2 | 14.19 | 0.65 |
| | | | 5,1,0 | ws | 2327.8 | 37.88 | <0.01 |
| | Coyhaique II | $PM_{2.5}$ | 5,1,0 | - | 1156.5 | 32.26 | <0.01 |
| | | | 4,1,0 | RH | 1114.6 | 3.81 | 0.09 |
| | | | 5,1,0 | $PM_{10}$ | 341.4 | 8.62 | <0.01 |
| | | | 5,1,0 | RH, $PM_{10}$ | 331.0 | 53.29 | <0.01 |
| | | | 5,1,0 | ws | 2327.8 | 37.88 | <0.01 |
| | | $PM_{10}$ | 1,1,2 | - | -308.5 | 4.14 | 0.35 |
| | | | 4,1,0 | RH | -297.4 | 38.41 | <0.01 |
| | | | 5,1,0 | $PM_{2.5}$ | -1056.2 | 48.60 | <0.01 |
| | | | 5,1,0 | RH, PM2.5 | -1040.6 | 49.90 | <0.01 |
| | | | 5,1,0 | ws | 894.6 | 31.14 | <0.01 |
| June | Ventanas | $SO_2$ | 4,1,3 | - | 12200.4 | 3.19 | 0.36 |
| | | | 3,1,4 | ws | 14198.5 | 5.43 | 0.14 |
| | | | 1,1,1 | ws, wd | 8897.2 | 7.00 | 0.22 |
| | | $PM_{2.5}$ | 1,1,2 | - | 8912.6 | 5.99 | 0.42 |
| | | | 1,1,2 | ws | -2115.8 | 9.89 | 0.05 |
| | | | 1,0,3 | ws, wd | 1432.8 | 19.13 | 0.07 |
| | Quintero | $SO_2$ | 1,0,1 | - | 14982.1 | 26.63 | <0.01 |
| | | | 5,0,2 | ws | 14963.4 | 4.62 | 0.20 |
| | | | 1,1,2 | ws, wd | 2656.6 | 6.54 | 0.37 |
| | | $PM_{2.5}$ | 3,1,5 | - | 9226.0 | 8.13 | 0.04 |
| | | | 3,1,3 | ws | 9217.0 | 5.43 | 0.14 |
| | | | 2,1,3 | ws, wd | 2030.6 | 4.69 | 0.32 |
| December | Ventanas | $SO_2$ | 5,1,3 | - | 14045.7 | 3.69 | 0.30 |
| | | | 2,1,3 | ws | 14037.6 | 6.33 | 0.18 |
| | | | 5,0,0 | ws, wd | 1981.8 | 16.49 | 0.09 |
| | | $PM_{2.5}$ | 1,1,2 | - | 10208.0 | 5.85 | 0.44 |
| | | | 1,1,2 | ws | 10196.7 | 6.55 | 0.26 |
| | | | 0,1,2 | ws, wd | -2479.3 | 23.58 | 0.05 |
| | Quintero | $SO_2$ | 2,0,2 | - | 17217.9 | 6.05 | 0.30 |
| | | | 1,0,2 | ws | 17236.6 | 39.54 | <0.01 |
| | | | 1,0,2 | ws, wd | 2892.3 | 7.53 | 0.18 |
| | | $PM_{2.5}$ | 3,1,5 | - | 10573.9 | 7.32 | 0.06 |
| | | | 3,1,3 | ws | 10560.6 | 4.49 | 0.21 |
| | | | 2,1,3 | ws, wd | 2287.7 | 4.58 | 0.33 |

RH: Relative Humidity, ws: wind speed, wd: wind direction

**Model analysis for Coyhaique.** The structural terms of the ARIMA model for monitoring station Coyhaique I and $PM_{2.5}$ showed an autoregressive term oscillating between 3 and 5. For the moving average term, the range was between 0 and 1. Finally, all the models have a value of 1 for the differencing term. All AIC values were less than 3000. The test using relative humidity as a covariable and without any external variables had lower Q-statistic values of 4.56 and 4.20, respectively, while the others had values greater than 30. Additionally, those models had p-values greater than 0.05, considered adequate [28].

After analyzing the residual graph at both stations for $PM_{2.5}$, only the test with relative humidity as a covariable showed the normality and white noise requirements. The rest exhibited lags outside the range, indicating significant autocorrelations. For this reason, the ARIMA (2,1,1) and (4,1,0) models achieved the best performances and were selected for the neural network analysis.

According to the analysis of the $PM_{10}$ concentration at the Coyhaique I station, models (3,0,0) and (1,0,0) yielded negative AIC values, contrasting with the other models. Notably, the model with wind speed as a covariable demonstrated a significantly greater AIC value of 2327.8. Considering the Ljung–Box statistic, only the lower-order model manifested a p-value of 0.65. Furthermore, a collective evaluation incorporating the Q statistic revealed consistent values, with the lowest observed value (14.19).

The autocorrelation function (ACF) showed the best performance for the (1,0,0) model, depicting a lack of significant autocorrelations across all lags. This configuration emerges as the most judicious choice for predicting $PM_{10}$ concentrations at the Coyhaique I monitoring station. This model was the most suitable candidate for predictions, corroborated by its adherence to white noise, indicating a normal distribution and optimal parameters (Guisande et al., 2011).

The structural terms of the identified models for the Coyhaique II station for the $PM_{2.5}$ pollutants were mostly the ARIMA (5,1,0) configuration. Moreover, the configuration with an autoregressive (AR) term of 4, a moving average (MA) term of 0, and a differencing parameter (d) of 1 exhibited the best performance. For the Ljung–Box test of the transformed models, the Q statistic was lowest at 3.81. Only the model (4,1,0) exhibited a p-value greater than 0.05, while the other models had p-values ranging from $10^{-6}$ to $10^{-11}$. A graphical examination of the autocorrelation function (ACF) revealed that models with relative humidity and wind speed as covariables were the only ones that displayed an appropriate structure for $PM_{2.5}$. In the case of residual plots, all the models exhibited a normal structure. As mentioned above, the ARIMA model (4,1,0) is suggested for predicting $PM_{2.5}$ at the Coyhaique II station.

The analysis of the $PM_{10}$ pollutant concentration at the Coyhaique II station revealed an ARIMA (1,1,2) structure without covariability. In contrast, the ARIMA (4,1,0) structure considered the relative humidity as an external variable, the same as that reported for $PM_{2.5}$. The remaining covariables were persistent with the ARIMA (5,1,0) configuration. Four negative values were obtained according to the AIC analysis of the transformed models. Moreover, in the Ljung–Box test, only one model exceeded the necessary value to consider white noise (p-value should be greater than 0.05): ARIMA (1,1,2).

Moreover, this configuration had the lowest Q statistic value, at 4.14, while the other values ranged between 30 and 63. Additionally, the graphs obtained for each model showed that only the model without covariables met the necessary lag range. However, all the configurations graphically exhibited white noise, and the residuals also demonstrated a normal distribution. Considering all the information, the ARIMA (1,1,2) model was deemed the most suitable for predicting $PM_{10}$. This evaluation considered which model best satisfied all the parameters assessed. For example, the AIC of the chosen model was not the lowest; however, other models lacked the appropriate p-value and did not exhibit the required graph adjustments. It is

important to remember that no single parameter alone was decisive in choosing the best model, so a comprehensive evaluation was necessary.

## Results of ANN models

The neural network models were built using the ARIMA model previously selected as a basis. The same covariates were considered for the model run for each monitoring station. Also, the Box-Cox method was used to treat the normality required.

The models had the structure (p,k); the first value shown (p) indicates how many observations of the same variable, but in previous periods, significantly affect the prediction, which was added as input variables, the second value (k) indicates the number of nodes or neurons that were in the hidden layer.

The model NNAR (1,2) resulted in both pollutants at Ventanas station during June, for $PM_{2.5}$ at the same location but in December, and only for $SO_2$ at Quintero station in both periods. It was the most repeated configuration. In those cases, the prediction only depends on observing the same variable from an earlier period. Also, there are two variables in the input layer, meaning there are only two neurons in this layer. The value of k equals two, indicating two nodes in the hidden layer, thus making it a simple but effective neural network. Since there were fewer neurons, there was better feedback between the nodes, and information and variables could be processed more accurately.

The resulting model for $PM_{2.5}$ at Quintero station during June and December was NNAR (2,2), indicating that the forecast depends on observations of the same variable from two periods before the one under study, thus having three neurons in the input layer to process the data. Concerning the neurons in the hidden layer, two nodes are present, similar to the discussion cases mentioned above, capable of effectively transferring and transforming information to the output layer.

For $SO_2$ at Ventanas station in December, the NNAR model (5.4) indicates that the predictions are affected by observations of the same variable of interest from five periods before. This model also points out that four neurons are within the hidden layer, which is to be expected due to a large number of inputs and a more extensive connection required to process all the incoming information.

In the case of the Coyhaique I monitoring station, the model showed, for $PM_{2.5}$ and using relative humidity as a covariate, an NNAR (2,2). Meanwhile, for Coyhaique II, it was NNAR (4,3). However, for the pollutant $PM_{10}$, using historical relative humidity information and $PM_{2.5}$ concentration as auxiliary variables, the resulting model was NNAR (2,1) for the Coyhaique I station. Meanwhile, the input for the Coyhaique II station was the historical $PM_{10}$ concentrations, resulting in NNAR (1,1).

## Performance of the ARIMA and the ANN models

Figs 4 and 5 show the time series comparison of the observed values and predicted registries obtained from ARIMA and ANN models.

The historical records for $PM_{2.5}$ and $PM_{10}$ during June at both stations had a wider range of values than the December registries. In other words, during the summer in Coyhaique, there are low oscillations of particulate matter compared to the winter period, when ventilation conditions have a marked influence on the dispersion of pollutants. Most predictions explain that more predicted points coincided with the observed values in December, showing a simulation with better statistics performance than those obtained in June. According to Fig 5, the last days of both months have more similar behaviors than the first 15 days. All models in their two comparison periods gave an $R^2$ statistic greater than 0.9, as shown in Table 2.

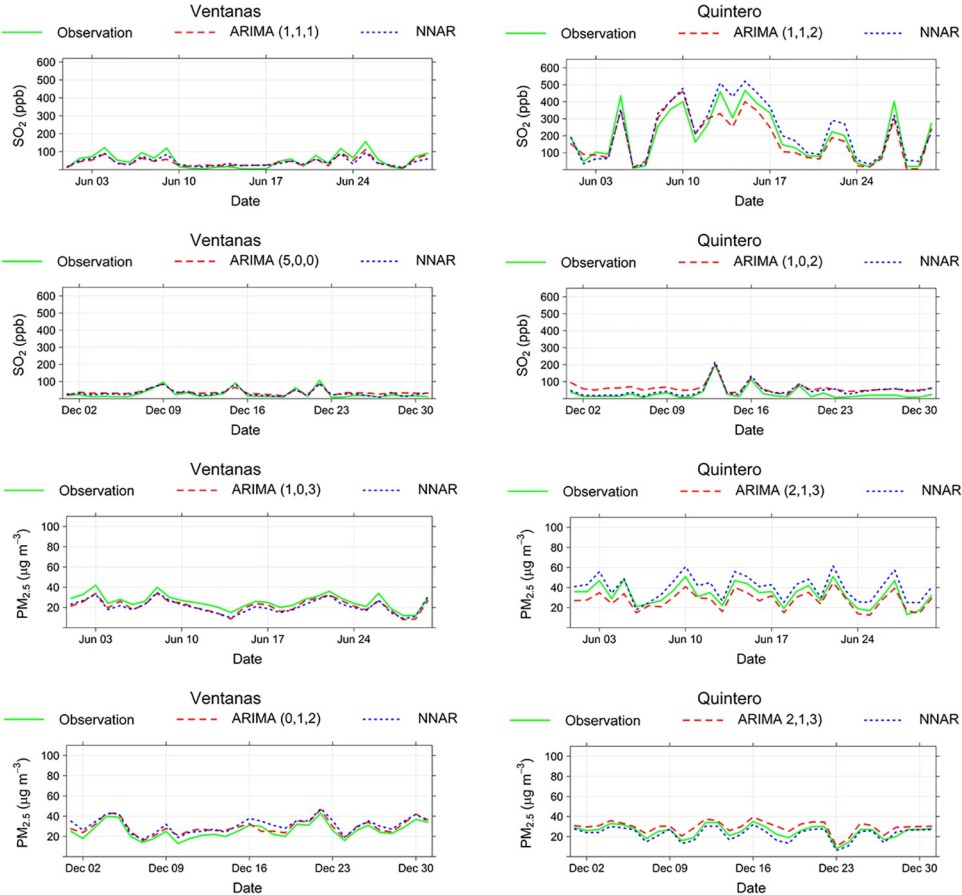

**Fig 4. Time series comparison for both models and the observed data in Ventanas and Quintero monitoring stations.**

In general, the predicted data in June 2019, both in neural networks and ARIMA models, differed from the observation in December 2019, which behaved more accurately in all models. If ARIMA and neural network models are compared, it is clear that the latter had a higher accuracy for both pollutants at the Coyhaique I and II monitoring stations. It is explained by its learning capabilities for self-adaptiveness to time series with fluctuating patterns and trends over time [28], which is the case with historical data in Coyhaique. Other studies have used many machine learning techniques for forecasting air quality. A scheme using an optimized recurrent neural network showed that the LSTM encoder-decoder model had the best performance and successfully forecasted $PM_{2.5}$ concentrations with a mean absolute percentage error (MAPE) of 28.2%, 15.07%, and 42.1% daily and 11.75%, 9.5%, and 7.4% hourly for different cities in Pakistan [50], similar values than achieved in this study.

On the other hand, using advanced technology, such as recurrent neural networks, contributes to the accuracy of air quality forecasting. These models facilitate the analysis of vast datasets, identifying intricate patterns and relationships among various variables [33], a capability shared with the hybrid ARIMA-ANN approach employed in the present investigation. The findings of this study align with those of prior research, such as [51], which demonstrated a similar performance of Artificial Neural Networks (ANNs). In that study, a unified architecture was employed, which combined tree-based architectures to capture spatial dependencies and temporal patterns. Using meteorological variables, air pollutant concentrations, and

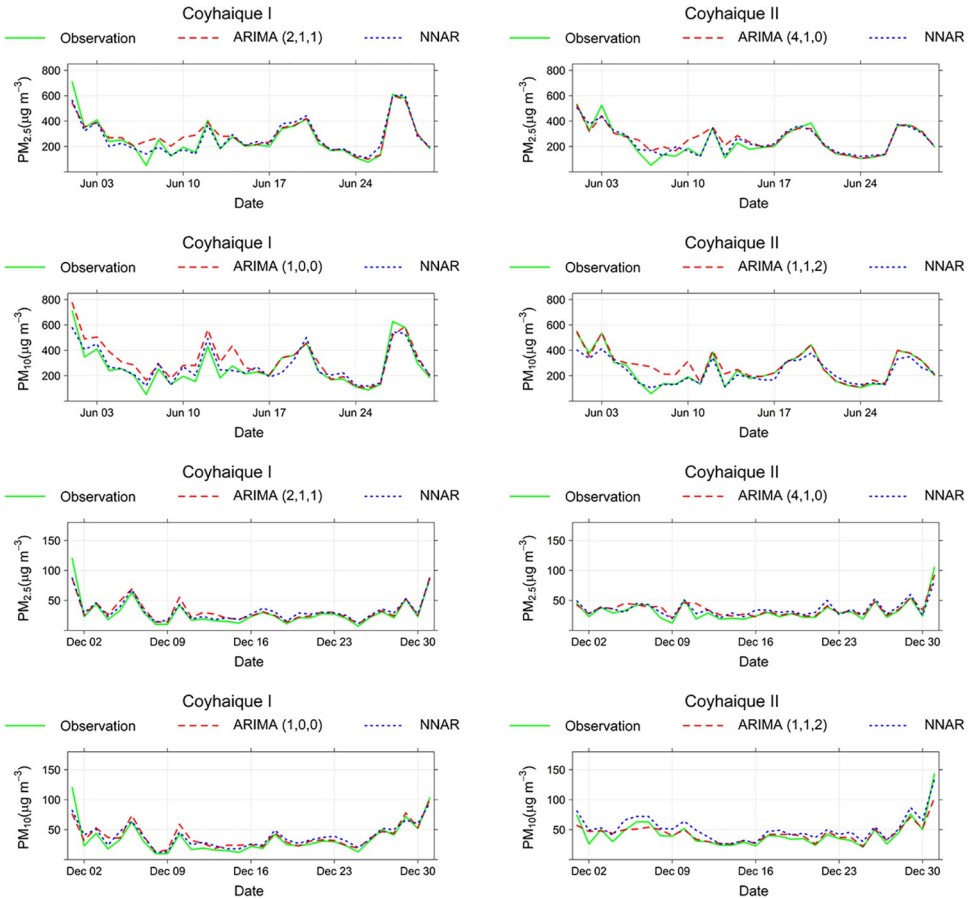

**Fig 5. Time series comparison for both models and the observed data in Coyhaique I and Coyhaique II monitoring stations.**

external data, it was trained to predict $NO_2$, $O_3$, $SO_2$, CO, $PM_{10}$, and $PM_{2.5}$ levels across diverse locations without retraining.

Wind direction significantly influences the transport and dispersion of pollutants from emission sources to monitoring stations. In industrial areas like Quintero and Puchuncaví, prevailing winds can carry pollutants toward residential zones, affecting air quality. By incorporating wind direction into the models, it becomes possible to account for these dynamics, leading to more accurate predictions of pollutant concentrations at specific locations. Similarly, wind speed affects the dilution and dispersion of pollutants. Higher wind speeds typically enhance dispersion, reducing pollutant concentrations, while lower wind speeds can lead to pollutant accumulation. Including wind speed as a covariate allows the models to adjust predictions based on these dispersion conditions, improving accuracy.

The improved performance metrics observed in the models that included external covariates underscore their importance. The models achieved higher $R^2$ values and exhibited lower RMSE and MAPE values, indicating more precise and reliable predictions. It demonstrates that integrating external meteorological factors enhances the models' ability to capture the complex interactions influencing air pollutant levels. In the case of the Coyhaique stations, the inclusion of relative humidity as an external covariate significantly improved model performance for PM₂.₅ and PM₁₀ predictions. The valley topography and climatic conditions in

**Table 2. Statistical results for ARIMA-ANN models.**

| Period | Monitoring station | Pollutant | ANN | | | ARIMA | | |
|---|---|---|---|---|---|---|---|---|
| | | | $R^2$ | RMSE | MAPE | $R^2$ | RMSE | MAPE |
| June | Coyhaique I | $PM_{2.5}$ | 0.94 | 41.67 | 0.26 | 0.88 | 60.35 | 0.26 |
| | | $PM_{10}$ | 0.90 | 53.08 | 0.30 | 0.86 | 78.82 | 0.29 |
| | Coyhaique II | $PM_{2.5}$ | 0.93 | 37.27 | 0.25 | 0.85 | 53.25 | 0.25 |
| | | $PM_{10}$ | 0.94 | 39.03 | 0.27 | 0.84 | 58.96 | 0.27 |
| December | Coyhaique I | $PM_{2.5}$ | 0.94 | 7.69 | 0.23 | 0.91 | 8.28 | 0.23 |
| | | $PM_{10}$ | 0.90 | 10.09 | 0.28 | 0.85 | 10.95 | 0.28 |
| | Coyhaique II | $PM_{2.5}$ | 0.91 | 5.84 | 0.22 | 0.85 | 7.76 | 0.22 |
| | | $PM_{10}$ | 0.94 | 10.60 | 0.14 | 0.90 | 10.07 | 0.14 |
| June | Ventanas | $SO_2$ | 0.90 | 22.08 | 0.90 | 0.89 | 21.34 | 0.92 |
| | | $PM_{2.5}$ | 0.93 | 5.13 | 0.20 | 0.92 | 4.52 | 0.17 |
| | Quintero | $SO_2$ | 0.92 | 51.93 | 0.38 | 0.91 | 49.93 | 0.30 |
| | | $PM_{2.5}$ | 0.93 | 7.65 | 0.25 | 0.92 | 6.68 | 0.19 |
| December | Ventanas | $SO_2$ | 0.91 | 10.50 | 0.56 | 0.91 | 15.34 | 0.89 |
| | | $PM_{2.5}$ | 0.92 | 5.68 | 0.23 | 0.91 | 4.15 | 0.17 |
| | Quintero | $SO_2$ | 0.90 | 22.26 | 1.29 | 0.89 | 35.15 | 2.31 |
| | | $PM_{2.5}$ | 0.92 | 3.04 | 0.12 | 0.90 | 4.81 | 0.20 |

Coyhaique contribute to pollutant accumulation, especially during winter when residential heating is prevalent.

Considering these insights, the predictive models developed in this study have significant implications for air quality management in Chile. The high accuracy of the hybrid ARIMA-ANN models underscores their potential as practical tools for forecasting pollutant concentrations like $SO_2$, $PM_{2.5}$, and $PM_{10}$. These models can inform early warning systems, enabling authorities to issue timely alerts and implement mitigation strategies to protect public health. Additionally, the models can guide policy decisions by highlighting the influence of meteorological factors and emission sources on air quality. Future investigations could focus on integrating additional contextual information specific to the analyzed regions, such as localized emission inventories, land use patterns, and socioeconomic factors. This integration could enhance the models' precision and adaptability, making them even more effective for region-specific applications. Expanding the modeling approach to include other pollutants and testing it in different geographical areas could further contribute to a comprehensive national strategy for air quality management. By leveraging advanced modeling techniques and incorporating a broader range of variables, these predictive models can be crucial in addressing environmental challenges and improving air quality forecasts in Chile.

## Conclusions

The comprehensive analysis of air quality data spanning the years 2016 to 2021 revealed critical insights into the atmospheric conditions of Chile, particularly in the cities of Quintero, Puchuncaví, and Coyhaique. Throughout the investigative period, applying a hybrid forecasting approach, integrating Autoregressive Integrated Moving Average (ARIMA) models and Artificial Neural Networks (ANN), emerged as a robust tool for predicting pollutant levels. Building upon fine-tuned ARIMA models, rigorously evaluated using metrics such as the Akaike Information Criterion (AIC) and Ljung-Box statistical tests, external covariates, including wind speed and direction, were incorporated to enhance model realism, especially within the Quintero industrial zone.

Meteorological analyses underscored the significant influence of geographical and climatic factors on air quality dynamics. In Coyhaique, the interplay of topography, seasonal wind patterns, and low temperatures during fall and winter created a stagnation effect, impeding the dispersion of pollutants. The study emphasized the importance of region-specific strategies for effective air quality management, acknowledging the unique environmental dynamics contributing to pollutant accumulation.

Evaluation of model performance consistently demonstrated the efficacy of the hybrid ARIMA-ANN approach. The models achieved $R^2$ values exceeding 0.90 across all monitored pollutants and stations, indicating strong correlations between predicted and observed values. Specifically, Mean Absolute Percentage Error (MAPE) values were below 1% for $PM_{2.5}$ predictions in Coyhaique, demonstrating exceptional model precision.

Comparisons with current literature trends, such as the increasing utilization of such models when larger datasets are available, underscored the relevance and effectiveness of the approach. Insights from recent studies, like those referenced, highlight the potential for further incorporating specific contextual information to enhance the accuracy of air quality predictions. The findings of this study significantly advance the understanding of air quality dynamics in Chile and advocate for the integration of advanced technologies, like neural networks, in future endeavors to improve air quality forecasting. By addressing the identified limitations—such as data availability and the need to incorporate additional influential variables—and exploring potential enhancements, researchers can further contribute to advancing this field, ultimately leading to better environmental management and public health outcomes.

## Author Contributions

**Conceptualization:** Fidel Vallejo.

**Data curation:** Fidel Vallejo, Nicolás Reinoso.

**Formal analysis:** Fidel Vallejo, Nicolás Reinoso, Luna Billartello.

**Funding acquisition:** Fidel Vallejo.

**Investigation:** Fidel Vallejo, Nicolás Reinoso, Luna Billartello.

**Methodology:** Fidel Vallejo, Nicolás Reinoso, Luna Billartello.

**Project administration:** Diana Yánez, Luis A. Díaz-Robles.

**Resources:** Diana Yánez, Luis A. Díaz-Robles, Luna Billartello.

**Software:** Luis A. Díaz-Robles, Marcelo Oyaneder, Lorena Espinoza-Pérez.

**Supervision:** Lorena Espinoza-Pérez, Ernesto Pino-Cortés.

**Validation:** Fidel Vallejo, Marcelo Oyaneder, Luna Billartello, Andrea Espinoza-Pérez, Ernesto Pino-Cortés.

**Visualization:** Fidel Vallejo, Patricia Viñán-Guerrero, Marcelo Oyaneder, Andrea Espinoza-Pérez, Lorena Espinoza-Pérez, Ernesto Pino-Cortés.

**Writing – original draft:** Fidel Vallejo, Diana Yánez, Patricia Viñán-Guerrero, Luis A. Díaz-Robles, Marcelo Oyaneder, Andrea Espinoza-Pérez, Lorena Espinoza-Pérez, Ernesto Pino-Cortés.

**Writing – review & editing:** Fidel Vallejo, Diana Yánez, Patricia Viñán-Guerrero, Andrea Espinoza-Pérez, Lorena Espinoza-Pérez, Ernesto Pino-Cortés.

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
