## [Decision Letter · Decision Letter 0]

26 Mar 2024

PONE-D-24-05815Enhancing Air Quality Predictions in Chile: Integrating ARIMA and Artificial Neural Network Models for Quintero and Coyhaique CitiesPLOS ONE

Dear Dr. Vallejo,

Thank you for submitting your manuscript to PLOS ONE. After careful consideration, we feel that it has merit but does not fully meet PLOS ONE’s publication criteria as it currently stands. Therefore, we invite you to submit a revised version of the manuscript that addresses the points raised during the review process.

**ACADEMIC EDITOR: Major Revisions.**

We look forward to receiving your revised manuscript.

Kind regards,

Worradorn Phairuang, Ph.D.

Academic Editor

PLOS ONE

Journal Requirements:

3. Please expand the acronym “DICYT” (as indicated in your financial disclosure) so that it states the name of your funders in full.

"No"

5. We note that your Data Availability Statement is currently as follows: [All relevant data are within the manuscript and its Supporting Information files.]

6. We note that [Figure 3] in your submission contain [map/satellite] images which may be copyrighted. All PLOS content is published under the Creative Commons Attribution License (CC BY 4.0), which means that the manuscript, images, and Supporting Information files will be freely available online, and any third party is permitted to access, download, copy, distribute, and use these materials in any way, even commercially, with proper attribution. For these reasons, we cannot publish previously copyrighted maps or satellite images created using proprietary data, such as Google software (Google Maps, Street View, and Earth). For more information, see our copyright guidelines: http://journals.plos.org/plosone/s/licenses-and-copyright.

a. You may seek permission from the original copyright holder of Figure 3 to publish the content specifically under the CC BY 4.0 license.  

Reviewers' comments:

Reviewer's Responses to Questions

**Comments to the Author**

1. Is the manuscript technically sound, and do the data support the conclusions?

Reviewer #1: Yes

Reviewer #2: Yes

2. Has the statistical analysis been performed appropriately and rigorously? 

Reviewer #1: Yes

Reviewer #2: Yes

3. Have the authors made all data underlying the findings in their manuscript fully available?

Reviewer #1: Yes

Reviewer #2: No

4. Is the manuscript presented in an intelligible fashion and written in standard English?

Reviewer #1: Yes

Reviewer #2: Yes

5. Review Comments to the Author

Reviewer #1: I have reviewed the manuscript "Enhancing Air Quality Predictions in Chile: Integrating ARIMA and Artificial Neural Network Models for Quintero and Coyhaique Cities" Before accept the manuscript kindly reply the below quires.

a) Why only two cities are chosen as sample sites ?

b) Kindly elaborate the methodology section of both ARIMA and ANN.

c) There are many models to predict the future phenomena like ANFIS, Tree models, optimization, AR, ARMA, MA, ML, AI, coupled etc models, why you select only these two models?

d) Conclusion section must be in point wise outcome of the current research.

e) Some more recent references should be added.

Reviewer #2: The contribution of "Integrated ARIMA and ANN models for accurate air quality predictions" lies in its approach of combining two well-established methods, AutoRegressive Integrated Moving Average (ARIMA) and Artificial Neural Networks (ANN), to improve air quality predictions.

1. Evaluation Metrics: It would be beneficial to employ a variety of evaluation metrics to assess the performance of the integrated ARIMA and ANN models comprehensively. Metrics such as mean absolute error (MAE), root mean square error (RMSE), and correlation coefficients can provide insights into different aspects of model performance.

2. Parameter Optimization: Fine-tuning the ARIMA and ANN models' parameters could improve prediction accuracy. Optimization techniques such as grid search or evolutionary algorithms can be employed to find the optimal set of parameters for the integrated model.

3. Feature Engineering: Exploring additional input features or incorporating domain knowledge into the modeling process may enhance the predictive capabilities of the integrated models. For example, meteorological data or data from other environmental sensors could be included to capture more comprehensive influences on air quality.

4. The novelty of the work depends on the specific context and the extent to which similar approaches have been explored in the literature. If integrating ARIMA and ANN models for air quality prediction is relatively uncommon or if the study introduces novel methodologies or insights, then the work can be considered novel. However, the novelty may be limited if similar integrated approaches have been extensively explored in the literature. A thorough literature review and comparison with existing studies can help determine the level of novelty in the work.

5. Potential Improvement: Spatially Explicit Modeling

The authors could consider employing spatially explicit modeling approaches to account for the heterogeneous distribution of pollutants influenced by Chile's unique topography. These methods allow for incorporating geographical features and terrain characteristics into air quality models, providing a more realistic representation of pollutant dispersion patterns. Additionally, integrating data from satellite imagery or remote sensing technologies can further enhance the spatial resolution of the modeling framework.

6. Potential Improvement: Hybrid Modeling Approaches

To address the limitations of the ARIMA model's reliance on autoregression and moving averages, the authors could explore hybrid modeling approaches that integrate multiple forecasting techniques. For example, ensemble methods combining ARIMA with machine learning algorithms such as ANN or gradient boosting machines (GBM) can leverage the strengths of each model while mitigating their weaknesses. By blending the predictive power of different methodologies, hybrid models may yield more robust and accurate air quality forecasts.

7. Comparison with the state-of the art literature is necessary, some of them are given as follows . https://ieeexplore.ieee.org/abstract/document/10363642 ; b. https://link.springer.com/article/10.1007/s11869-015-0369-9; c.https://www.mdpi.com/2071-1050/13/21/12217; d. https://sensors.myu-group.co.jp/sm_pdf/SM3362.pdf

e. https://ieeexplore.ieee.org/abstract/document/10288482 .

While the integrated ARIMA and ANN models represent a valuable contribution to air quality prediction, addressing the identified limitations and exploring potential enhancements can further advance the research in this field. By incorporating spatially explicit modeling techniques and exploring hybrid modeling approaches, researchers can strive to improve the accuracy and reliability of air quality forecasts, ultimately contributing to better environmental management and public health outcomes.

6. PLOS authors have the option to publish the peer review history of their article (what does this mean?). If published, this will include your full peer review and any attached files.

Reviewer #1: **Yes: **Dr. Kulwinder Singh Parmar

Reviewer #2: **Yes: **SHUBHANKAR MAJUMDAR

---

## [Author Response · Author response to Decision Letter 0]

7 May 2024

Dear Academic Editor and Reviewers,

The attached document details the changes made to address the observations and suggestions provided for the manuscript. We deeply appreciate the time you have dedicated to reviewing the document.

Best regards,

Dr. Fidel Vallejo

---

## [Decision Letter · Decision Letter 1]

22 May 2024

PONE-D-24-05815R1Enhancing Air Quality Predictions in Chile: Integrating ARIMA and Artificial Neural Network Models for Quintero and Coyhaique CitiesPLOS ONE

Dear Dr. Vallejo,

Thank you for submitting your manuscript to PLOS ONE. After careful consideration, we feel that it has merit but does not fully meet PLOS ONE’s publication criteria as it currently stands. Therefore, we invite you to submit a revised version of the manuscript that addresses the points raised during the review process.

**ACADEMIC EDITOR: Major Revision**

We look forward to receiving your revised manuscript.

Kind regards,

Worradorn Phairuang, Ph.D.

Academic Editor

PLOS ONE

Reviewers' comments:

Reviewer's Responses to Questions

**Comments to the Author**

1. If the authors have adequately addressed your comments raised in a previous round of review and you feel that this manuscript is now acceptable for publication, you may indicate that here to bypass the “Comments to the Author” section, enter your conflict of interest statement in the “Confidential to Editor” section, and submit your "Accept" recommendation.

Reviewer #1: All comments have been addressed

Reviewer #2: (No Response)

2. Is the manuscript technically sound, and do the data support the conclusions?

Reviewer #1: Yes

Reviewer #2: Partly

3. Has the statistical analysis been performed appropriately and rigorously? 

Reviewer #1: Yes

Reviewer #2: No

4. Have the authors made all data underlying the findings in their manuscript fully available?

Reviewer #1: Yes

Reviewer #2: No

5. Is the manuscript presented in an intelligible fashion and written in standard English?

Reviewer #1: Yes

Reviewer #2: Yes

6. Review Comments to the Author

Reviewer #1: accepted, all queries are addressed well. Revised manuscript is well written and statistical notations are now correct updated.

Reviewer #2: The authors should highlight the edits, unable to recognize in which area the author has updated in the manuscript. In broader sense, applying existing solutions to a real problem does not offer any technical contribution, thus the technical contribution and novelty of the solution is very limited, the author should provide a framework development like - https://arxiv.org/abs/2404.05482 ; https://ieeexplore.ieee.org/document/10529960 which can shows that your results are better.

7. PLOS authors have the option to publish the peer review history of their article (what does this mean?). If published, this will include your full peer review and any attached files.

Reviewer #1: No

Reviewer #2: **Yes: **Shubhankar Majumdar

---

## [Author Response · Author response to Decision Letter 1]

22 May 2024

Dears Worradorn Phairuang, Ph.D. and Shubhankar Majumdar,

I hope this message finds you well.

Reviewer 2 has indicated that the following points have not been adequately addressed or completed: 1) Statistical analysis, 2) Complete availability of data, 3) Visibility of changes in the manuscript, and 4) Clarification of the novelty of the study. 

In response, the authors would like to address these concerns as follows:

1. The manuscript has been carefully revised according to the guidelines provided by both reviewers. We have been informed that Reviewer 1 has already given their approval for the publication of this manuscript. Regarding point 1, the statistical analysis, including model selection, analysis, validation, and the metrics and criteria used, is clearly detailed in Section 2: Methodology, from lines 215 to 275. It is worth noting that in the first round, a favorable opinion was given on this point by both reviewers.

2. Concerning point 2, it is stated in the response letter, page 2, that the data and the R script are fully publicly available on the Kaggle repository, with DOI: 10.34740/kaggle/ds/4949501.

3. Regarding point 3, we submitted two versions of the manuscript to the journal: the document with changes and the document without changes. Below are the specific lines where changes have been made: lines 59, 64, 164, 177, 211, 225-249, 275, 446, 467, 483-494, 499-505, 512-522. In some cases, changes have even been highlighted in yellow for greater visibility. Each change is mentioned in the response letter.

4. Finally, the novelty of the study is articulated in the manuscript and demonstrated through the hybrid approach used for two different pollutants in two areas with specific geography and economic activities. This is achieved by applying hybrid ARIMA models with artificial neural networks. The justification for the study is provided in lines 151-157, and a comparison with results available in the literature is established in lines 472-494. The conclusions have been rewritten considering the incorporation of sources provided by the reviewer in the first round.

We hope these clarifications adequately address the concerns raised by Reviewer 2. 

Thank you for your attention and consideration.

Best regards,

Fidel Vallejo, PhD. 

National University of Ecuador

---

## [Decision Letter · Decision Letter 2]

6 Jun 2024

PONE-D-24-05815R2Enhancing Air Quality Predictions in Chile: Integrating ARIMA and Artificial Neural Network Models for Quintero and Coyhaique CitiesPLOS ONE

Dear Dr. Vallejo,

Thank you for submitting your manuscript to PLOS ONE. After careful consideration, we feel that it has merit but does not fully meet PLOS ONE’s publication criteria as it currently stands. Therefore, we invite you to submit a revised version of the manuscript that addresses the points raised during the review process.

**Please consider responding to the reviewer.**

We look forward to receiving your revised manuscript.

Kind regards,

Worradorn Phairuang, Ph.D.

Academic Editor

PLOS ONE

Reviewers' comments:

Reviewer's Responses to Questions

**Comments to the Author**

1. If the authors have adequately addressed your comments raised in a previous round of review and you feel that this manuscript is now acceptable for publication, you may indicate that here to bypass the “Comments to the Author” section, enter your conflict of interest statement in the “Confidential to Editor” section, and submit your "Accept" recommendation.

Reviewer #2: (No Response)

2. Is the manuscript technically sound, and do the data support the conclusions?

Reviewer #2: Partly

3. Has the statistical analysis been performed appropriately and rigorously? 

Reviewer #2: No

4. Have the authors made all data underlying the findings in their manuscript fully available?

Reviewer #2: Yes

5. Is the manuscript presented in an intelligible fashion and written in standard English?

Reviewer #2: Yes

6. Review Comments to the Author

**Reviewer #2: **No proper framework has been developed.

There is a lack of proper justification in the manuscript that how the number of neurons, hidden layers of ANN is taken while integrating ARIMA.

There is a lack of optimization of parameters, found in the the manuscript.

Comparison of the results should be done by newer publications.

Could find proper response to the previous comments, require proper justification.

There is nothing written about the overfitting of the models.

How many epochs are required for the simulation of ANN model is not written anywhere in the manuscript.

7. PLOS authors have the option to publish the peer review history of their article (what does this mean?). If published, this will include your full peer review and any attached files.

Reviewer #2: **Yes: **Dr. Shubhankar Majumdar

---

## [Author Response · Author response to Decision Letter 2]

8 Jul 2024

Responses to Reviewer 2's comments are in the attached document.

Please view the document as having tracked changes to view the additions of text and paragraphs that have been modified.

---

## [Decision Letter · Decision Letter 3]

8 Oct 2024

PONE-D-24-05815R3Enhancing Air Quality Predictions in Chile: Integrating ARIMA and Artificial Neural Network Models for Quintero and Coyhaique CitiesPLOS ONE

Dear Dr. Vallejo,

Thank you for submitting your manuscript to PLOS ONE. After careful consideration, we feel that it has merit but does not fully meet PLOS ONE’s publication criteria as it currently stands. Therefore, we invite you to submit a revised version of the manuscript that addresses the points raised during the review process.

Please revise slightly to incorporate the new requests. No need to satisfy all the comments.

We look forward to receiving your revised manuscript.

Kind regards,

Yangyang Xu

Academic Editor

PLOS ONE

Journal Requirements:

Reviewers' comments:

Reviewer's Responses to Questions

**Comments to the Author**

1. If the authors have adequately addressed your comments raised in a previous round of review and you feel that this manuscript is now acceptable for publication, you may indicate that here to bypass the “Comments to the Author” section, enter your conflict of interest statement in the “Confidential to Editor” section, and submit your "Accept" recommendation.

Reviewer #2: (No Response)

Reviewer #3: (No Response)

2. Is the manuscript technically sound, and do the data support the conclusions?

Reviewer #2: Partly

Reviewer #3: Yes

3. Has the statistical analysis been performed appropriately and rigorously? 

Reviewer #2: No

Reviewer #3: Yes

4. Have the authors made all data underlying the findings in their manuscript fully available?

Reviewer #2: No

Reviewer #3: Yes

5. Is the manuscript presented in an intelligible fashion and written in standard English?

Reviewer #2: No

Reviewer #3: Yes

6. Review Comments to the Author

Reviewer #2: Some suggestions:

1. Enhance the clarity of the abstract by providing a brief overview of the methodology and key findings.

2. Include a section on the limitations of the study to provide a comprehensive understanding of the research scope and potential areas for future research.

3. Provide a detailed explanation of the data collection process, including the selection criteria for monitoring stations and the frequency of data acquisition.

4. Discuss the implications of the research findings for air quality management in Chile, emphasizing the practical applications of the predictive models.

5. Add newer literature like https://ieeexplore.ieee.org/abstract/document/10529960 and compare your result with the article.

6. Consider expanding the discussion on the significance of external covariates like wind speed and direction in improving prediction accuracy.

7. Incorporate a subsection on the methodology detailing the steps involved in integrating ARIMA models with Artificial Neural Networks for forecasting air quality parameters.

8. Include a glossary of key terms to assist readers in understanding technical terminology used in the paper.

9. Provide a summary of the key challenges faced during model development and evaluation, highlighting the strategies employed to address them.

10. Consider adding a section on the societal impact of air pollution in Quintero, Puchuncaví, and Coyhaique to contextualize the importance of the study.

Reviewer #3: Overall evaluation:

(1) Main contribution:

The authors focus on air quality predictions, employing a hybrid forecasting strategy (ARIMA, ANN). Their research is conducted in Chile (Quintero, Ventanas, Coyhaique I, and Coyhaique II). Model performance is evaluated based on metrics (AIC, Ljung-Box statistical tests, and diverse statistical indicators). The hybrid ARIMA-ANN models, enriched with external covariates like wind speed and direction, led to good predictive prowess (R2 > 0.90). The dataset is significant, spanning five years (2016 – 2021). The source is the National Air Quality Information System from Chile (219 monitoring stations for meteorological variables and air quality).

(2) Defects of the paper:

(a) The frequency of the data in your dataset for all analysed pollutants is not mentioned. For the sake of the reader, it should be added. Can you say something about the accuracy of the monitoring systems used by your National Air Quality Information System from Chile?

(b) Some words are exaggerated (“demonstrated unparalleled predictive prowess”). I know papers that apply hybrid algorithms of ML found a better R2. Indeed, they are not for Chile, but the idea is the same.

(c) This study's limitations are not well emphasized. The same idea applies to the novelty elements.

(d) A short description of the studied area and the climate from there is missing.

(e) Conclusions are not clear enough at this moment. My suggestion is to use numbers for objectivity.

(3) Minor mistakes:

Figures 4, 5, and 6 do not have good resolution, but they might be connected to the PDF. Please check.

(4) Questions:

(a) The author states: “Over the period from 2016 to 2021, the data available for PM2.5 and PM10 at the Coyhaique I monitoring station stood at 98.2% and 97.9%, respectively, while for Coyhaique II, these figures were 96.5% for PM2.5 and 96.0% for PM10. On the other hand, Quintero stations showed between 96.2% and 98.8 % of data availability.” Do you suggest that there is missing data in the National Air Quality Information System from Chile? If the answer is positive, how many of them are in those time intervals chosen by you during Summer and Winter?

(b) Why did you choose to analyse only PM2.5, PM10 and SO2 as pollutants?

(c) Why do you think that the proposed models perform differently in different locations?

(d) How did you divide the dataset into training and validating datasets?

(5) Innovation elements: The discussions regarding the new findings can be improved. I suggest comparing their findings with other worldwide studies to emphasise the novelty elements. The authors provided an analysis of the potential of advanced technologies in refining air quality forecasts in Chile.

(6) Overall recommendation: With the identified issues addressed, the paper has the potential to be accepted after minor revisions. The authors are encouraged to consider the feedback provided and make the necessary adjustments to improve the clarity, significance, and objectivity of their paper.

7. PLOS authors have the option to publish the peer review history of their article (what does this mean?). If published, this will include your full peer review and any attached files.

Reviewer #2: No

Reviewer #3: **Yes: **Mihaela Tinca Udristioiu

---

## [Author Response · Author response to Decision Letter 3]

17 Oct 2024

Dear Reviewers and Editor,

We sincerely appreciate the feedback provided and have carefully addressed all comments. The suggested revisions have been incorporated into the attached file.

---

## [Editor Report · Decision Letter 4]

8 Nov 2024

Enhancing Air Quality Predictions in Chile: Integrating ARIMA and Artificial Neural Network Models for Quintero and Coyhaique Cities

PONE-D-24-05815R4

Dear Dr. Vallejo,

We’re pleased to inform you that your manuscript has been judged scientifically suitable for publication and will be formally accepted for publication once it meets all outstanding technical requirements.

Kind regards,

Yangyang Xu

Academic Editor

PLOS ONE
---

## [Editor Report · Acceptance letter]

12 Nov 2024

PONE-D-24-05815R4 

PLOS ONE

Dear Dr. Vallejo, 

I'm pleased to inform you that your manuscript has been deemed suitable for publication in PLOS ONE. Congratulations! Your manuscript is now being handed over to our production team.

Kind regards, 

on behalf of

Dr. Yangyang Xu 

Academic Editor

PLOS ONE